# Maxillary Sinusitis as a Complication of Zygomatic Implants Placement: A Narrative Review

**Riccardo Nocini** [1,*], **Giorgio Panozzo** [2], **Alessandro Trotolo** [2] and **Luca Sacchetto** [1]

[1] Department of Otorhinolaryngology—Head and Neck Surgery, University Hospital Verona, 37100 Verona, Italy; luca.sacchetto@univr.it

[2] Department of Maxillofacial Surgery and Dentistry, University Hospital Verona, 37100 Verona, Italy; panozzo.g27@gmail.com (G.P.); ale.trotolo@gmail.com (A.T.)

[*] Correspondence: riccardo.nocini@gmail.com

**Abstract:** Aims: The aim of this review is to consider maxillary sinusitis as a complication of zygomatic implants placements. Maxillary sinusitis a common complication but in the literature there are no reviews that focus only on this condition and its possible treatment. This review was carried out with to highlight the main findings of the literature on this topic and to improve knowledge in this field. Methods: The search strategy resulted in 155 papers. After selection of the inclusion criteria only 11 papers were examined. From the papers these, 12.3% patients presented maxillary sinusitis but only four studies evaluated sinusitis (both clinical and radiological evaluation). The most common treatment used by the authors were antibiotics alone or combined with functional endoscopic sinus surgery (FESS). Results: The literature shows an absence of precise and shared guidelines diagnosis and post-operative follow-up, and of the treatment of maxillary sinusitis following zygomatic implantology. It has not been determined if the surgical placement of ZIs is better than the other techniques for treatment of the onset of maxillary sinusitis in the post-operative period. Conclusion: To date there are no shared protocols for maxillary sinusitis treatment. In our experience, and according to the literature in the presence of risk factors such as age, comorbidities, smoking, nasal septal deviation or other anatomical variants, we suggested that FESS is performed at the same time as placement of zygomatic implants.

**Keywords:** zygomatic implants; sinusitis; FESS; maxillofacial surgery; endoscopic surgery; otorhinolaryngology





## 1. Introduction

Implant-based dental rehabilitation can be achieved with multiple approaches. The use of conventional implants requires adequate bone height and width in patients. In patients with severe atrophic maxilla (classes V e VI according to the Cawood-Howell classification of edentulous jaw) conventional rehabilitation is not a viable option. In this scenario, Brånemark et al. [1] proposed positioning zygomatic implants (ZIs) as a reliable technique for maxillary rehabilitation using the resistant zygomatic bone as an anatomical site for anchoring [2]. Various other alternatives are commonly employed in maxillary atrophy, such as sinus floor elevation for bone grafting, free re-vascularized flaps, Le Fort I osteotomy and bone grafts from the iliac crest or other sites [3–5].

Different techniques have been proposed for ZI placement, and many authors have contributed to improve and modify Brånemark's initial technical proposal [6] in which ZIs were placed within an intrasinusal pathway using a sinus window to elevate and preserve the sinus membrane. Nowadays, the zygomatic anatomy guided approach (ZAGA) proposed by Aparicio [7] is gaining greater success, in which the surgical technique is customized to the patient's anatomy in relationship to the maxillary sinus. The extrasinusal approach involves placing the implants outside the maxillary sinus without compromising the Schneider's membrane, while the sinus slot technique proposed by Stella et al.

in 2000 [8] is a technique involving consideration of extrasinusal and intrasinusal positioning [6]. Nowadays, the most used protocol for the edentulous maxilla involves the placement of two zygomatic implants in the premolar area and two to four conventional dental implants in the premaxilla area to provide additional support for a full-arch maxillary prosthesis [9]. Furthermore, the QUAD-ZYGOMA technique describes the use of four ZIs (two for each side) in cases where the premaxilla does not allow the placement of conventional implants [10].

Regardless of the surgical technique, successful outcomes have been demonstrated. Chrcanovic et al. in 2016 showed a cumulative survival rate of 95% at 12 years, with most failures occurring in the first 6 months after surgery [11]. Generally, the level of patient satisfaction was excellent as reported by Sartori and colleagues in 2012 [12].

Even though prosthetic failure is infrequent, ZIs are not free of complications. Maxillary sinusitis is reported to be the most common [13–15] and, together with intraoral soft-tissue infection [2,7,15], chronic pain, nerve deficits [13,16,17] and oro-antral fistula [18,19], may reduce implant survival and undermine patient's quality of life.

While many studies have focused on long-term implant stability, less importance has been dedicated to describing how ZIs can affect maxillary sinuses [20], as well as the management and/or prevention of sinusitis.

To date, maxillary sinusitis is known to be the most frequent complication after placement of zygomatic implants. However, there are no studies in the literature that focus only on this complication and highlight the modalities of onset, diagnosis, and the principles of treatment. The purpose of this study is to review the literature to assist the surgeon in managing the possible occurrence of post-operative maxillary sinusitis after zygomatic implant placement.

## 2. Research Method

This review was carried out with a narrative approach to highlight the main findings of the literature concerning the specific topic, and to outline and improve knowledge in this field [21]. Specifically, the approach of by Egger et al. was pursued to perform a narrative review of the literature according to the following steps [22] (Table 1).

**Table 1.** Methodological approach to review.

| Step | General Activities | Specific Activities |
|------|-------------------|---------------------|
| I | Formation of working group | Two otolaryngologists selected as experts in head and neck anatomy and sinusitis surgery, as clinical and methodological operators. |
| | | Two maxillofacial surgeons selected as expert in zygomatic implant surgery and in maxillary sinus anatomy, as clinical and methodological operators. |
| II | Formulation of the review questions | Evaluation of the state of art treatment of sinusitis, and complications following zygomatic implant surgery. Analysis of main complications of zygomatic implant surgery and their treatment |
| III | Identification of relevant studies on PubMed and PMC | 1. Identification of keywords in the field of interest |
| | | 2. Use of Boolean operator (AND) |
| | | 3. Inclusion criteria: no time limitation for works published; language: English; all types of full text articles. Exclusion criteria: no full text available, reviews of the literature and reviews of the literature with meta-analyses |
| | | 4. Elimination of duplicates |
| | | 5. Manual search through the references in selected articles |
| IV | Anaysis and presentation | Extrapolation of data from all revised studies and their presentation in the form of a narrative review |

Step I: formation of a four-member working group, each member a clinical expert acting as a methodological reviewer.

Step II: formulation of review questions derived from up-to-date findings on zygomatic implants surgery, surgical outcomes, complications and treatments. This is required to effectively build a search strategy.

Step III: using keywords concerning the topic to perform searches in databases (PubMed, PMC, Ovid, Scopus). Boolean operators as well as the Mesh terms were not applied to minimize potential restrictions in finding results (Table 2).

**Table 2.** Search strategy.

| Search | Database |
|---|---|
| (zygomatic implants) AND (maxillary sinusitis) | PMC, PubMed, Ovid Scopus |
| (zygomatic surgery) AND (maxillary sinusitis) | PMC, PubMed, Ovid, Scopus |
| (zygomatic surgery) AND (sinusitis) | PMC, PubMed, Ovid, Scopus |
| (intrasinusal technique zygomatic implants) AND (maxillary sinusitis) | PMC, PubMed, Ovid, Scopus |
| (extrasinusal technique zygomatic implants) AND (maxillary sinusitis) | PMC, PubMed, Ovid, Scopus |
| (extrasinus zygomatic implants) AND (maxillary sinusitis) | PMC, PubMed, Ovid, Scopus |
| (extrasinus zygomatic implants) AND (complications) | PMC, PubMed, Ovid, Scopus |
| (intrasinusal zygomatic implants) AND (complications) | PMC, PubMed, Ovid, Scopus |

Inclusion article criteria were original articles, related only to humans, without limitation of publication date, and no limitation concerning the study design was applied. Exclusion article criteria were articles for which the full text was not available, were not in English, or were grey literature. Reviews and meta-analyses were excluded.

Duplicate articles were eliminated and, from the articles retrieved in the first round of the search, additional references were identified by a manual search among the cited references. The process of literature selection was reported following the PRISMA statement guidelines and is clarified in Figure 1.

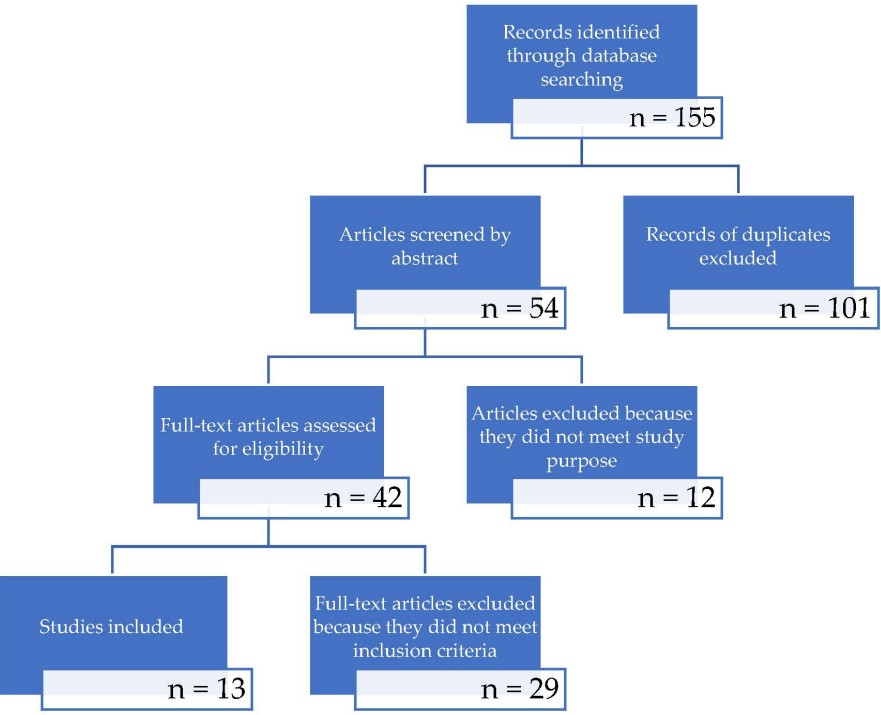

**Figure 1.** Study screening process.

To clarify the topics of this review the outcomes were identified as follows:

- Presence of maxillary sinusitis (clinical and/or radiological signs and symptoms).
- Correlation between sinusitis and extra or intra-sinusal surgical technique.
- Need for pharmacological and/or surgical treatment of sinusitis.
- Presence of protocol for the treatment of the patients.

The endpoint ranged between 1 and 3 years after surgery.

Step IV: analysis of all data extrapolated was carried out by authors and data presented in the form of a narrative review, as in Table 3 in the results section of this paper.

Works were excluded in which preoperative symptoms were described without proper interpretation of diagnosis. Works in which patients presented preoperative sinusitis were excluded so that postoperative results were not compromised (selection bias).

The type of prosthetic load of the implants was not considered as this is irrelevant in the development of sinusitis [23–25]. It was important to highlight any correlations between surgical positioning techniques and the development of sinusitis, as well as related surgical treatments.

The period in which the selected studies were published ranged from 1998 and November 2021.

The PICOTS method was followed:

- **P: Participants**. Adult patients that underwent ZI placement to rehabilitate the superior dental arch in the case of maxillary atrophy (Cawood-Howell V–VI) alone or combined with conventional implants.

  None of the patients presented signs and/or symptoms of sinusitis at the time of surgery.

- **I: Intervention**. Zygomatic implant placement (all surgical techniques were considered in the study)
- **C: Comparison control**. Not applicable
- **O: Outcomes**. Presence of maxillary sinusitis; clinical and/or radiological signs and symptoms; correlation between sinusitis and extra or intra-sinusal surgical technique; need for pharmacological and/or surgical treatment of sinusitis
- **T: Time.** Between 1 year and 3 year after intervention
- **S: Studies**. All clinical studies were included. Reviews and meta-analyses were excluded.

## 3. Results

### 3.1. Literature Research

The search strategy resulted in 155 papers found between April and May 2021. Once the duplicates were excluded, 54 articles were considered. The abstracts were independently screened by the authors searching for articles related to the focus question, resulting in 42 studies.

After reading the full text of the resulting studies, 29 articles were excluded because they did not meet the inclusion criteria or did not match the outcomes of this paper. Thus, a total of 13 publications were included in the review (Table 3).

### 3.2. Description of the Studies and Analysis

Eleven of the included studies were retrospective [26–36], while two were prospective studies [14,37]. Study year of publication ranged from 2008 to 2021. The papers included 478 patients and 1156 zygomatic implants overall. All of the patients were reported to be free of clinical sinusitis in the pre-operative evaluation. All the patients had a minimum follow-up period of 12 months and a maximum registered follow-up of 144 months (12 years). Nine authors used an extrasinusal surgical technique (classical or sinus slot technique). One article described the ZAGA technique [32] and two presented both intrasinusal and extrasinusal approach [33,34]. Bothur et al. [36] chose the intrasinusal technique. A total of 107 ZIs were placed with an intrasinusal approach, and 765 with an extrasinusal approach (including those positioned with a sinus slot technique). Finally, Perla Della Nave et al. [32] positioned 102 ZIs with the Zygoma anatomy guided approach (ZAGA). Surgical technique was not

clarified by the authors for 141 ZIs. During post-operative evaluation, 69 patients (14.5%) presented sinusitis, of which 42 (8.8%) had clinical relevance. Only five of the studies reported clinical and radiological evaluation separately [27,29,34,36,37]. D'Agostino et al. [37] assessed post-operative clinical sinusitis with a SNOT-20 questionnaire and radiological signs with the Lund-McKay scale, observing a statistically significant increasement of radiologic sinusitis, which was not followed by significant clinical variations.

Only patients that reported clinical evidence of post-operative sinusitis were treated by the authors: 18 patients were treated with antibiotic alone while 11 patients received medical treatment combined with functional endoscopic sinus surgery (FESS). Two patients underwent Caldwell-Luc surgery. Seven patients needed partial (one patient) or complete (six patients) ZI removal to relieve symptoms.

In one patient treated with combined antibiotic and FESS, clinical recurrence of sinusitis was observed. No other recurrences were registered after the chosen treatment.

**Table 3.** Articles considered in the review.

| Author | Paper Design | Year | N° of Patients | Mean Age | Risk Factors for Sinus Pathologies | N° of Zygomatic Implants | Pre-op Sinusitis | Pre-op Imaging | Post-op Sinusitis | Post-op Imaging | Post-op Symptoms | Treatment | Results after Treatment | Follow-up | Surgical Technique |
|---|---|---|---|---|---|---|---|---|---|---|---|---|---|---|---|
| Urgell et al. [26] | Retrospective | 2008 | 54 | 56 (38–75) | Smokers (8/54 < 10 sigg, 1/54 > 10 sigg). 1/54 HIV + | 101 | 0/54 | HRCT + OPG | 1.9% (1/54) | OPG | N.A. | Antibiotics | Regression (1/1) | 72 months | Extrasinusal |
| Davò et al. [27] | Retrospective | 2008 | 42 | 57 (34–79) | Smokers < 10 sigg | 81 | 0/42 | Yes (not specified) | 2.3% (intra-op oro-antral communication) (1/42) | OPG + A-P Teleradiograph | Fluid drainage from nasal to oral cavity | Antibiotics + FESS antrostomy | Regression (1/1) | 20.5 (12–42) months | Extrasinusal, sinus slot, minimally invasive |
| Davò et al. [35] | Retrospective | 2008 | 36 | N.A. | Smokers (4/36 < 10 sigg) | 71 | 0/36 | CT paranasal sinuses | 0% clinical, 5.6% radiologic (2/36) | CT paranasal sinuses | No symptoms | Follow-up | No treatment | 21.9 (13–42) months | Extrasinusal (61/71), sinus slot (10/71) |
| Davò et al. [28] | Retrospective | 2009 | 24 | 51.4 (36–72) | Smokers | 45 | 0/24 | N.A. | 20.8% (5/24) | OPG + A-P teleradiograph + CT (5/24) | N.A. | Antibiotics (2/5), FESS (2/5), Caldwell-Luc approach (1/5) | Regression (5/5) | 60 months | Extrasinusal |
| Chow et al. [14] | Prospective | 2010 | 16 | 60 (no range) | Smokers (2/16) | 37 | No | CBCT | 0% | CBCT | No symptoms | Follow-up | No treatment | 12 months | Extrasinusal |
| Aparicio et al. [29] | Retrospective | 2012 | 22 | 63 (48–80) | Smokers (17/22 nonsmokers, 2/22 11–20 sigg, 3/22 > 20 sigg) | 41 | 0/22 | CBCT | 27.3% clinical (6/22), 54.6% radiologic (12/22) | CBCT | According to Lanza-Kennedy scale | Antibiotics + antihistamine (5/6), partial implant removal (1/6) | Regression (4/6) | 120 months | Extrasinusal |
| Bothur er al. [36] | Retrospective | 2015 | 14 | 60 (51–78) | Smokers (3/14), allergic rhinitis (1/14) | 58 | 0/14 | Standard X-rays | 28.5% (4/14 clinical)100% (14/14 radiological signs | CBCT | Yes (self-administered questionnaire) | Antibiotic | Regression (47) | 117.6 months | Extrasinusal |
| Bertolai et al. [30] | Retrospective | 2015 | 31 | 62 (52–82) | Smokers (0/31), Diabetes (0/31) | 78 | 0/31 | CT + OPG | 6.5% (2/31) | CT (in patients w. Clinical sinusitis) | N.A. | FESS 1/31, FESS + Caldwell-Luc approach 1/31 | Regression (2/2) | 20–60 months | Extrasinusal (27/31), Extra-maxillary (4/31) |

**Table 3.** *Cont.*

| Author | Paper Design | Year | N° of Patients | Mean Age | Risk Factors for Sinus Pathologies | N° of Zygomatic Implants | Pre-op Sinusitis | Pre-op Imaging | Post-op Sinusitis | Post-op Imaging | Post-op Symptoms | Treatment | Results after Treatment | Follow-up | Surgical Technique |
|---|---|---|---|---|---|---|---|---|---|---|---|---|---|---|---|
| Araujo et al. [31] | Retrospective | 2017 | 37 | 55.64 | N.A. | 129 | 0/37 | CBCT | 21.62% (8/37) | CBCT, OPG | Congestion, cough, purulent nasal drainage, chronic pain | Antibiotics 5/37, Antibiotics + FESS 3/37 | Regression (7/8), Recurrence (1/8) | >12 months | Sinus Slot |
| D'Agostino et al. [37] | Prospective | 2019 | 13 | 58 | No risk factors | 52 | 0/13 clinical, 0/26 sinuses w. Radiologic LMS = 0 | CBCT | 23% (3/13)/11.5% (3/26) of sinuses w. Radiologic LMS 1 | CBCT | No symptoms | FESS | SNOT20 1.2 (average) | 45 months | Extrasinusal |
| Perla Della Nave et al. [32] | Retrospective | 2020 | 102 | N.A. | N.A. | 206 | 0/102 | N.A. | 4.9% (5/102) | OPG, periapical radiographs, CBCT | N.A. | Antibiotics, conservative approach, implant removal | Regression after 8 months of antibiotic therapy (1/8), spontaneous after less than 1y. (1/8), implant removal after 2 y (1/8)–5 y (1/8)–10 y (1/8) | 12–144 months | ZAGA |
| Yalcin et al. [33] | Retrospective | 2020 | 45 | 51.76 | N.A. | 141 | 0/45 | CBCT | 4.44% (2/45) | OPG (+CT/CBCT) | N.A. | Removal (3/141 ZI) | N.A. | 6–36 months | Intrasinusal, extrasinusal, sinus slot |
| D'Agostino et al. [38] | Retrospective | 2021 | 42 | 63 (29–81) | Smokers (8/42) | 116 | 0/42 clinical, 13/84 sinuses w. radiologic LMS > 1 | CT | 19% clinical 8/42) (29/82 sinuses with radiologic LMS > 1) | CT | Yes (not specified) | Antibiotic and/or FESS | Regression (8/8) | 60 (12–162) months | Intrasinusal (32) Extrasinusal (10) |

## 4. Discussion

Zygomatic implants (ZIs) represent one of the most used therapeutical choices in response to severe atrophic maxillae (Cawood-Howell V and VI) rehabilitation. There are different techniques to place ZIs. Parel et al. [39] used a crestal incision to displace a large flap facilitating the exposure of the zygomatic bone, followed by the creation of a window for the displacement of the sinus membrane. The technique of Stella [8] differed from the original technique, as there was no need for a classical window opening on the wall of the maxillary sinus. Instead, they proposed a slot acting as a smaller antrostomy to orient the twist drill for implant placement. The third technique is completely performed outside the maxillary sinus, without windows opening or channels in the wall of the maxillary sinus. This has been used for patients with a pronounced maxillary sinus lateral wall concavity [40].

The most used ZI positioning protocol for edentulous maxilla usually involves two zygomatic implants in the premolar area and two to four standard dental implants placed anteriorly to provide additional support for a full-arch maxillary prosthesis [9]. Furthermore, the Quad technique provides the use of four ZIs, two for each side, where the premaxilla does not allow positioning of traditional implants. This protocol can support the whole prosthesis. Stiévenart and Malevez [15] evaluated patients treated with the Quad approach and reported that the cumulative survival rate after 40 months was 96%; generally; the weighted average success in zygomatic implant positioning is 97.05% [5].

In comparison to conventional treatment, ZIs preserve the patient from multiple surgical procedures and reduce the entire rehabilitation process (immediate loading is also described), time for healing and the risk of infections. Moreover, use of ZIs avoid extraoral harvesting for osseointegration, providing predictable success and absolute retention. Even if it prosthetic failure was rare, some authors [11,41–43] demonstrated some drawbacks related to ZIs. They are positioned closed to delicate structures in the midface (e.g., maxillary and palatine artery, facial and V2 nerve branches), and because of this the surgical procedures should be performed by a trained surgeon. In addition, any problems related to ZIs are more difficult to approach than in conventional dental implant rehabilitation. During the early stages after surgery, sinusitis appears to be one of the most common complications.

The 13 publications included in this review evaluated 478 patients and 1156 zygomatic implants overall, with only 69 patients (14.5%) with sinusitis, of which 42 (8.8%) showed clinical relevance in the follow up.

From a diagnostic point of view, maxillary sinusitis may be observed radiologically (usually by orthopantomography or, more often, maxillofacial computed tomography with clinical assessment. These diagnostic tools are useful not only for proper evaluation and surgical planning before ZI placement, but for monitoring the clinical and prosthetic outcomes in more detail afterwards. Otherwise, in the literature examined, there was no evidence of precise diagnostic analysis of sinusitis itself.

From this review, clinical outcomes of sinusitis seemed difficult to find when compared with radiological outcomes [27], which is a critical point considering the greater evidence of radiological signs [7] that can cause silent sinusitis with otorhinolaryngological consequences, both chronically and over time [44,45]. Indeed, both symptomatic and silent sinusitis were taken into account in this review to extend the analysis to the evaluation of sinusitis diagnosis (nasal endoscopy and radiological assessment) and treatment (FESS vs. simple antibiotic vs. follow-up).

The literature to date does not specifically report on the status of the region of the osteo-meatal complex (COM) and the middle meatus. Particularly, there is a lack of the following information:

1. Alterations of the natural ostium of the maxillary sinus.
2. Presence of the accessory ostium, which could modify the physiological drainage capacity from the maxillary sinus as well as (in the case of surgical treatment )modify the approach to setting up at typical antrostomy.

3. Alteration of the middle meatus with consequent difficulty of sinus drainage due to a paradoxical curvature of the middle turbinate or its pneumatization with a concha bullosa, which, in both cases, would drastically reduce the space of the middle meatus resulting in the condition of dysventilation.

The above should be taken into account and considered as further correlated risk factors for sinusitis onset after ZI positioning [35,46,47].

Preserving the sinus membrane is still debated, and no evidence in literature could be found to establish a strict protocol to follow for implant zygomatic surgery. However, damage of the maxillary mucosa due to a protruding implant is not the leading cause in developing sinusitis and not linked to a shorten survival rate of ZI [24,48]. Nevertheless it may be advisable to avoid oroantral communication during zygomatic implant placement. Chow et al. [14] suggested an extrasinusal procedure, in which a retained bone window is made up and used as a shield to preserve the attached sinus membrane intact before positioning the implant. Avoiding the breach of the membrane by the extrasinusal position of the ZI leads to positive results and perfect osteointegration. Galli et al. in 2001 [49] explained this by the absence of mobility of these implants, avoiding irritation of the sinus mucosa and/or obstruction of the meatal complex. However, seven patients needed partial (one patient) or complete (six patients) ZI removal to relieve symptoms of sinusitis. Many factors were associated with the onset of sinusitis, from a blockage of the maxillary ostium [50] to the poor osseointegration of the implants in the posterior palatal region undermining prosthetic stability [18] or the creation of a direct link for the passage of bacteria from the oral cavity into the maxillary sinus [15]. In addition, trauma involved in the placement of zygomatic implants can cause a maxillary hemosinus which, by slowing down the epithelial and mucociliary activity in the sinus, can contribute to the onset of sinusitis [51].

In the literature reviewed there was no definite and clear evidence about the relationship between maxillary sinusitis and the technique used for ZI positioning. The great majority of surgeons seems to prefer the extrasinusal approach (765 ZIs with an extrasinusal approach compared to 107 ZIs with an intrasinusal approach), but a detailed analysis of an increase in complications in terms of sinusitis of one technique compared to the other was not possible. It emerged that there are episodes of sinusitis even in cases of extrasinusal implants placement. In this case, therefore, one possible goal might be to evaluate, both from a clinical and radiological point of view, the timing of the onset of sinusitis to assess the pathogenesis of sinusitis. It may be possible to find reactive sinusitis six months after the surgery [52] specifically related to surgical procedures.

In this regard, it could be interesting to investigate, in the case of the extra-sinusal technique, where the onset of sinusitis is not immediately correlated to the surgical maneuvers, the timing of the FESS treatment.

In the cases analyzed in this review, regardless of the surgical technique used the treatment of sinusitis was approached with administration of antibiotic therapy in most cases. This needs further study, since antibiotic therapy alone may resolve the acute condition but not completely resolve the pathology of the sinus. However, no univocal treatment protocol emerged. This raises the question of the need for surgical treatment by FESS in favor of a single antibiotic therapy with further follow-up on signs and symptoms (with published references) for the treatment of sinusitis arising after zygomatic implant surgery. Further study is required on the analysis of associated risk factors to establish a prophylactic protocol beforehand, and to identify surgical or conservative responses later, for these cases of sinusitis. Unfortunately, the heterogeneity of the studies and the diagnostic and instrument tools used did not allow for a consistent result concerning this topic. In some cases, risk factors were found to have no relevance in facilitating sinusitis; however, this could not be stated conclusively because of lack of information about diagnosis of the disease and its assessment during the follow-up. For example, smoking and an average age over 52 years could be a predisposing factor for the onset of maxillary sinusitis with the need for close clinical and radiological follow-up.

Considering this, the authors believe that the main focus has to be on the diagnostic protocol (radiological, probably) to gain sound information. Starting from here, the association between risk factors, surgical techniques and the onset of sinusitis could be clear, and advice in terms of follow-up and therapeutic strategies could be provided with confidence. Based on the results of this review, an otolaryngological approach should be combined with surgical and prosthetic rehabilitation, due to the occurrence of sinusitis after these kinds of procedures. Presumably, the otolaryngologist would be involved in the diagnostic and follow-up phases, as well as during the surgical procedure when an FESS is appropriate.

Given the onset of sinusitis following the placement of ZIs, it may be appropriate to evaluate the feasibility of intraoperative FESS intervention, requiring interventions under general anesthesia.

The authors are well aware of the difficulty in designing in vivo clinical studies, especially in heterogeneous cases like those of zygomatic implant surgery; however, performing a diagnosis of the status of maxillary sinus and monitor it during follow-up using instrumental tools is highly recommended. This allows the gathering of unambiguous information. Based on such information, a more predictable protocol could be established for treatment of sinusitis after zygomatic surgery and the potential need for referral to an otorhinolaryngologist.

We observed that 8.8% of patients presented maxillary sinusitis with clinical relevance that needed pharmacological or surgical treatment. However, to date, there are no standardized therapeutic protocols described in the literature. In the analyzed papers, we found a heterogeneous approach in treating maxillary sinusitis. Some authors perform FESS in a contextual manner to prevent the onset of sinusitis, while others perform it only in the event of the onset of maxillary sinusitis. In most of the cases, antibiotic treatment was the prime choice.

The present study has some limitations due to the nature of the literature reviewed. Most were clinical studies and data about diagnosis, radiology, surgical techniques and therapies were not often reported; this lack of information has to be considered. Because of this, a narrative review was carried out and it was impossible to determine exact guidelines for management of maxillary sinusitis following zygomatic implant surgery. Clinical conditions, the patients' health status and variables related to surgical procedures hindered the establishment of clear and sound guidelines in terms of management and therapy of this peculiar complication. Furthermore, maxillary sinusitis, the focus of the present work, often occurs with other side effects from this kind of surgery, and because of this it is impossible to establish strict protocols.

## 5. Conclusions

The literature shows an absence of precise and shared guidelines on the diagnosis, post-operative follow-up and treatment of maxillary sinusitis following zygomatic implantology, despite the incidence of this complication, with clinical symptoms and radiological incidence up to 54%.

Furthermore, it is not yet defined if and how different ZI placement and surgical techniques influence the onset of maxillary sinusitis in the post-operative period.

In the absence of risk factors, a follow-up that includes the execution of a CBCT 6 months after the placement of the implants, with possible FESS in case of radiological signs and/or symptoms of sinusitis, is recommended.

Antibiotic medical therapy should always be administered to avoid maxillary silent or clinically relevant sinusitis, even after many months.

In the presence of risk factors such as age, comorbidities, smoking, nasal septal deviation and other anatomical variants, FESS should be performed simultaneously with placement of the zygomatic implants.

The advantages of zygomatic implant placement in atrophic maxillae far outweigh the risk of developing maxillary sinusitis. In light of the results of the present review, it is

concluded that when this complication occurs it can be successfully treated with medical therapy alone, or in combination with FESS, and has little influence on implant survival.

A shared protocol is still necessary to aid surgeons in therapeutical choices when post-operative sinusitis occurs.

**Author Contributions:** Conceptualization, R.N. and L.S.; methodology, R.N.; validation, G.P. and A.T.; formal analysis, G.P.; investigation, R.N.; material and method, G.P.; data curation, A.T.; writing—original draft preparation, R.N.; writing—review and editing, R.N.; supervision, L.S. All authors have read and agreed to the published version of the manuscript.

**Funding:** This research received no external funding.

**Institutional Review Board Statement:** Not applicable.

**Informed Consent Statement:** Informed consent was obtained from all subjects involved in the study. Written informed consent has been obtained from the patient to publish this paper.

**Data Availability Statement:** This study did not report any data.

**Conflicts of Interest:** The authors declare no conflict of interest.

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
