# Peer review of "Maxillary Sinusitis as a Complication of Zygomatic Implants Placement: A Narrative Review"

_applsci, doi:10.3390/app12020789_

Round 1

Reviewer 1 Report

The topic sounds original and with an interesting clinical meaning. However, there are some aspects to be clarified or modified:

  1. I would ask you why do you use a narrative review if the title refers to a systematic review?
  2. The Abstract should be structured in subcategories: Aims, methods, results and conclusions
  3. In the Research Method, instead of table 1 I believe a Flow Chart of the search method and strategy would be much more appropriate.
  4. In the Results section, table 3 should be placed there, and the title of each article included in the search should be deleted, as it is not relevant for the readers.
  5. In the Discussion section, line 132- "32 of 55" refers to what? Also, ENT refers to what? Maybe all the acronyms should be explained several times across the manuscript, so the text would be easier to be followed.
  6. Also, maybe the Discussion section should be somehow structured so it follows the four outcomes mentioned in the Research method: "Presence of maxillary sinusitis (clinical and/or radiological signs and symptoms)
    • Needing of pharmacological and/or surgical treatment of sinusitis
    • (Correlation between sinusitis and) extra or intra-sinusal surgical technique
    • Presence of protocol for the treatment of the patients"
  7. In the Conclusion section, any citations are not appropriate
  8. It would also be very interesting to emphasize better the advantages of using zygomatic implants in oral surgery and why the risk of developing maxillary sinusitis is lower than the advantages of using them.

Author Response

Dear reviewer,

Thank you for your suggestions. All corrections were incorporated in the text when needed and all modifications were red-highlighted.

1)I would ask you why do you use a narrative review if the title refers to a systematic review? 

1) An editing mistake was done, thank you for the advice, the title was modified.

2)The Abstract should be structured in subcategories: Aims, methods, results and conclusions

2) The abstract was re-edited in subcategories as you suggested.

3) In the Research Method, instead of table 1 I believe a Flow Chart of the search method and strategy would be much more appropriate.

3) Table 1 was maintained to better explain the different steps of a narrative review, furthermore, the Flow Chart of the search method was added.

4) In the Results section, table 3 should be placed there, and the title of each article included in the search should be deleted, as it is not relevant for the readers.

4) The table 3 was moved in the Results section and the title of each article was deleted

5)In the Discussion section, line 132- "32 of 55" refers to what? Also, ENT refers to what? Maybe all the acronyms should be explained several times across the manuscript, so the text would be easier to be followed.

5) In the Discussion section, "32 of 55" is referred to the patients treated with FESS (Functional Endoscopic Sinus Surgery) and/or antibiotics. All the acronyms were explained, ENT means Ears Nose Throat (synonymous of Otolaryngologists). Clarifications were reported in the text.

6)Also, maybe the Discussion section should be somehow structured so it follows the four outcomes mentioned in the Research method:

• Presence of maxillary sinusitis (clinical and/or radiological signs and symptoms)
• Needing of pharmacological and/or surgical treatment of sinusitis 
• (Correlation between sinusitis and) extra or intra-sinusal surgical technique 
• Presence of protocol for the treatment of the patients"

6) The discussion was modified as you suggested following the four points mentioned in the Research method

7)In the Conclusion section, any citations are not appropriate

7) Some references were edited.

8) It would also be very interesting to emphasize better the advantages of using zygomatic implants in oral surgery and why the risk of developing maxillary sinusitis is lower than the advantages of using them.

8) The aim of the present study was exactly to emphasize better the advantages of using zygomatic implants in oral surgery and why the risk of developing maxillary sinusitis is lower than the advantages of using them. Indeed, our findings clearly highlights the feasibility of using this kind of surgery because the complications are almost of minimal importance and, when they occur, it is possible to deal with them with different kind of therapies, both surgical (FESS) and medical (antibiotic drugs). A sentence was added to discussions.

Reviewer 2 Report

Title

  • There is a grammatical error in the title, what do you mean by “complicance”, do you mean complication. The authors should revise the title to exactly reflect the contents and aims of study.

 Abstract

  • There is disharmony between the background and the aim of the study. The aim is unclear, what do the authors mean by “preoperative studies”?, the authors should state briefly in one statement rationale of study before mentioning aim of study.
  • Why the authors submit the abstract without methods, results, and conclusion?

Introduction

  • What are the implications, contributions, and novelty of the present review? Knowing that the following systematic reviews on the same topic were published (https://pubmed.ncbi.nlm.nih.gov/?term=zygomatic+implants&filter=pubt.systematicreview)

Methods

  • The type of the review as written in the title is “systematic review” but the authors stated that “This review was carried out with a narrative approach that can allow highlighting main findings of literature about the specific topic to outline and to improve knowledge in this field. Specifically, concepts proposed by Egger et.al were pursued to perform narrative review of the literature according to the following steps”. The authors should clearly state clearly the type of review they are conducted (Systematic or narrative review). If this review was systematic review as the author stated earlier in the title, the authors should adhere to the PRISMA criteria for conducting systematic review (Page MJ, McKenzie JE, Bossuyt PM, et al. The PRISMA 2020 statement: an updated guideline for reporting systematic reviews. BMJ 2021;372:n71. doi:10.1136/bmj.n71).
  • What was the focus question of the current study?
  • In the Search strategy, the authors didn't determine the period in which the selected studies were published.
  • The protocol of the current review was not registered in PROSPERO platform; however, protocol registration is not mandatory in several Journals.
  • The search strategy was limited to the PubMed and this not enough. The authors should conduct search strategy in another database too (Embase, Scopus, Ovid, Cochrane library……etc).
  • How risk of bias was assessed?
  • What data was collected from the included studies?
  • The inclusion and exclusion criteria were unclear, the authors should follow PICOTS criteria. P: Type of participants (healthy adult patient with maxillary atrophy (classification of bone atrophy), smokers ……etc; what about patient with history of sinusitis and thickening of Schneiderian membrane. I: Patient receiving zygomatic implant, C: if applicable, patient receiving conventional implants with lateral sinus lift. O: outcomes was sinusitis …..etc. T: time, S: type of included studies. The authors should state the minimum number of the patients to be considered when selecting study for inclusion.
  • The authors should provide more details about the search terms used in the search strategy. Please add Mesh terms in your search strategy.

Results

  • The authors should provide more details about how many studies yielded at different screening processes.
  • The authors should include PRISMA Flowchart diagram.
  • In general, the results were not appropriately reported.  
  • Several studies evaluating maxillary sinusitis following zygomatic implant placement were not included, example:

For example: 1-Chow J, Wat P, Hui E, Lee P, Li W. A new method to eliminate the risk of maxillary sinusitis with zygomatic implants. Int J Oral Maxillofac Implants. 2010 Nov-Dec;25(6):1233-40. PMID: 21197502.

2- Bothur S, Kullendorff B, Olsson-Sandin G. Asymptomatic chronic rhinosinusitis and osteitis in patients treated with multiple zygomatic implants: a long-term radiographic follow-up. Int J Oral Maxillofac Implants. 2015 Jan-Feb;30(1):161-8. doi: 10.11607/jomi.3581. PMID: 25615923.

If the above mentioned studies were not included due to any reasons, the author should provide more details about the excluded studies and reasons for exclusion.

Discussion

  • What is the novelty of the current systematic review as compared to the currently available reviews? Several systematic reviews were published, please see the link below https://pubmed.ncbi.nlm.nih.gov/?term=zygomatic+implants&filter=pubt.systematicreview
  • What are the limitations of the current study? this should be written in the discussion along with justification. Also, what are the recommended solution for such challenges and their issues?  

Author Response

Dear reviewer,

Thank you for your suggestions. All corrections were incorporated in the text when needed and all modifications were red-highlighted.

Title. An editing mistake in the title was done, thank you for the advice, the title was modified.

Abstract. The abstract was rearranged in the different sections.

Introduction. Modifications were incorporated in the text

Method.

1)period of publication. The period evaluated in the search was reported. Authors selected studies published between 1998 and November 2021. This information was  added in the text, as suggested.

2)Prospero. This is a narrative review. Unfortunately, PROSPERO accepts only protocols for registering systematic reviews of humans and animals studies, as it could be seen in its site (https://www.crd.york.ac.uk/prospero/).

3)The search strategy. The authors omitted to mention the other databases consulted because the results did not change in terms of paper resulted and therefore selected for the review. Moreover, as it could be seen in table 1 of this paper, https://www.ncbi.nlm.nih.gov/pmc/articles/PMC4504929/, narrative reviews differs from systematic reviews in terms of inclusion of studies and search methods that are based on authors’ intuition and research experience for the former and find out mainly in PubMed or Medline database for the latter. However, following your suggestion, also other databases were mentioned in the material and methods section.

4)Risk of bias. Usually, risk of bias is assessed by “risk of bias assessment tool” and this instrument takes into account allocation concealment and blinding, etc. However, in the present narrative review authors did not measured risk of bias, just discussed limitations of different studies included. This choice was related to the nature of our work.

5)Data collected. Data collected were reported in Table 3, however data analyzed for the aim of the review (outcomes) are reported in “material and methods” section.

6)PICOTS criteria. PICOTS criteria are reported in the text, in the material and methods section.

7) Mesh terms. Mesh terms are restrictive and very circumstantial, they are designed to make detailed searches. This tool is appropriate for focusing the search. To focusing the results is not the aim of a narrative review, where the results have to be as inclusive as possible. For this reason, authors did not taken into consideration Mesh terms.

Results:

Prisma flow chart was added as you requested.

Results was modified inclusion and exclusion criteria clarified.

1-Chow J, Wat P, Hui E, Lee P, Li W. A new method to eliminate the risk of maxillary sinusitis with zygomatic implants. Int J Oral Maxillofac Implants. 2010 Nov-Dec;25(6):1233-40. PMID: 21197502.

Was esclude because after the abstract reading the paper talk about the method of placement and not about the treatment of a possible maxillary sinusitis after the placement f ZI.

2- Bothur S, Kullendorff B, Olsson-Sandin G. Asymptomatic chronic rhinosinusitis and osteitis in patients treated with multiple zygomatic implants: a long-term radiographic follow-up. Int J Oral Maxillofac Implants. 2015 Jan-Feb;30(1):161-8. doi: 10.11607/jomi.3581. PMID: 25615923.

After your suggestion we reviewed and added to our papers the study of Bothur.

We clarify the inclusion and exclusion criteria of all the articles examined in the full text reading.

Discussion:

Thank you for your suggestion, actually, the up-to-date literature well highlights the complications of zygomatic surgery and implant rehabilitation. The aim of the present study was to find sound results only about complications in terms of onset of maxillary sinusitis. We are well aware that literature provides satisfying results about the zygomatic implant surgery and all complications, however, we aimed to find out some information about sinusitis and its treatment: this was done with the purpose to establish some clinical advice to follow by clinicians dealing with this pathology after zygomatic implant surgery.

Round 2

Reviewer 1 Report

The manuscript has been improved

Author Response

The manuscript has been improved

Thanks for your appreciation and for your suggestion in the first round of revision

Reviewer 2 Report

 Abstract

  • Aim: The authors stated that “The aim of this review is to take in consideration the maxillary sinusitis as a complication of zygomatic implants placements. It’s a common complication otherwise in litterature there are not any review focused only on this topic and the possible treatment”. However, there were some systematic reviews published previously:

1- https://pubmed.ncbi.nlm.nih.gov/22562293/  

2- https://pubmed.ncbi.nlm.nih.gov/33600519/

3- https://pubmed.ncbi.nlm.nih.gov/34209770/

  • Methods: The authors should briefly Specify the information sources (e.g. databases) used to identify studies, the date (From ....to …) when each was last searched, study design (experimental and/or observational studies), outcomes of interests, methods used to present and synthesis the results.

The method section in Revision 1 should be in the Results section not in the Methods section.

  • Grammar errors such as 1-line 15 litterature should be literature, 2- line 20, were take should be “were taken”.
  • Conclusion: the conclusion is unrelated to the aim of the study.

Introduction

  • Knowing that the following systematic reviews on the same topic were published

1- https://pubmed.ncbi.nlm.nih.gov/22562293/  ; 2-https://pubmed.ncbi.nlm.nih.gov/33600519/ 3- https://pubmed.ncbi.nlm.nih.gov/34209770/

Also, knowing that Systematic review and meta-analysis are at the top of the of evidence pyramid, what are the implications, contributions, and novelty of the narrative review as compared to previously published systematic review?

Methods

  • I think this topic is better to be addressed with systematic review. The authors should adhere to the PRISMA criteria for conducting systematic review (Page MJ, McKenzie JE, Bossuyt PM, et al. The PRISMA 2020 statement: an updated guideline for reporting systematic reviews. BMJ 2021;372:n71. doi:10.1136/bmj.n71).
  • What was the focus question of the current study?
  • In the Search strategy, the authors should provide Supplementary table reporting details search strategy along with number of yielded studies from each database.

Results

  • Several studies evaluating maxillary sinusitis following zygomatic implant placement were not included. As part of search strategy, the author should check the references of the previously published studies. Please read this systematic review as there are several studies reporting maxillary sinusitis after zygomatic implant and were ignored by the authors” Chrcanovic BR, Abreu MH. Survival and complications of zygomatic implants: a systematic review. Oral Maxillofac Surg. 2013 Jun;17(2):81-93. doi: 10.1007/s10006-012-0331-z. Epub 2012 May 6. PMID: 22562293.”

Discussion

  • What are the limitations of the current study? this should be written in the discussion along with justification.

Author Response

Abstract 

  • Aim: The authors stated that “The aim of this review is to take in consideration the maxillary sinusitis as a complication of zygomatic implants placements. It’s a common complication otherwise in litterature there are not any review focused only on this topic and the possible treatment”. However, there were some systematic reviews published previously:

1- https://pubmed.ncbi.nlm.nih.gov/22562293/  

2- https://pubmed.ncbi.nlm.nih.gov/33600519/

3- https://pubmed.ncbi.nlm.nih.gov/34209770/

The reviews cited are focused on all the complications of zygomatic implant surgery. They are all very interesting and we used them to analyze the topic. However, our work is focused only on the maxillary sinusitis following the zygomatic implant placement and not on all possible complications, therefore we assume that the present narrative review is not an overlapping of other reserches.

  • Methods: The authors should briefly Specify the information sources (e.g. databases) used to identify studies, the date (From ....to …) when each was last searched, study design (experimental and/or observational studies), outcomes of interests, methods used to present and synthesis the results

The databases  are reported in table 2, as suggested in the previous round of review. Furthermore, the study designs of all studies analyzed are reported in the table 3, where authors showed and explained the results considered.

  • Grammar errors such as 1-line 15 litterature should be literature, 2- line 20, were take should be “were taken”.

The grammar errors were checked

  • Conclusion: the conclusion is unrelated to the aim of the study.

In our opinion, conclusions clearly report that nowadays there are no protocols in literature useful to be shared and commonly used in the clinical routinely practice of zygomatic surgery to avoid or to prevent maxillary sinusitis.

Introduction

  • Knowing that the following systematic reviews on the same topic were published

1- https://pubmed.ncbi.nlm.nih.gov/22562293/  ; 2-https://pubmed.ncbi.nlm.nih.gov/33600519/3- https://pubmed.ncbi.nlm.nih.gov/34209770/

Also, knowing that Systematic review and meta-analysis are at the top of the of evidence pyramid, what are the implications, contributions, and novelty of the narrative review as compared to previously published systematic review?

The aim of our narrative review was to analyze only maxillary sinusitis following zygomatic implant surgery. We are well aware that meta-analyses and systematic reviews are stronger in evidences, however, we did not find in the up-to-date literature systematic reviews analyzing only maxillary sinusitis as complication of zygomatic implant surgery and the present work claim to give the readers new perspectives and proposals to prevent and manage this specific complication.

Methods 

  • I think this topic is better to be addressed with systematic review. The authors should adhere to the PRISMA criteria for conducting systematic review (Page MJ, McKenzie JE, Bossuyt PM, et al.The PRISMA 2020 statement: an updated guideline for reporting systematic reviews. BMJ2021;372:n71. doi:10.1136/bmj.n71).
  • What was the focus question of the current study?

Finding in the literature a correlation of maxillary sinusitis and  zygomatic implant surgery, finding some shared protocols or guidelines (there are not) and  proposing some clinical advice or guideline to prevent and manage properly this complication.

  • In the Search strategy, the authors should provide Supplementary table reporting details search strategy along with number of yielded studies from each database.

The value of review does not consist in complicating information, in our opinion inserting numbers of papers found and discarded will lead to confusion and reading difficulties without a real improvement of the validity of the methodology of the work

  • Results
  • Several studies evaluating maxillary sinusitis following zygomatic implant placement were not included. As part of search strategy, the author should check the references of the previously published studies. Please read this systematic review as there are several studies reporting maxillary sinusitis after zygomatic implant and were ignored by the authors” Chrcanovic BR, Abreu MH. Survival and complications of zygomatic implants: a systematic review. Oral Maxillofac Surg. 2013 Jun;17(2):81-93. doi: 10.1007/s10006-012-0331-z. Epub 2012 May 6. PMID: 22562293.”

Thank you for the suggestion, all studies included answered inclusion criteria of the review chosen by authors during the conceptualization. Other studies were discarded.

Discussion 

  • What are the limitations of the current study? this should be written in the discussion along with justification.

The present study has some limitation due to the nature of works in literature. Indeed, they are mostly clinical studies and data about diagnosis, radiology, surgical techniques and therapies are not often all reported and lack of information has to be considered. Precisely for this, a narrative review was carried out and it was impossible to give exact guidelines to manage maxillary sinusitis following zygomatic implant surgery. Clinical conditions, patients’ health status and variables related to surgical procedures hinder to establish clear and sound guidelines in terms of management and therapy of this peculiar complication. Furthermore, maxillary sinusitis, focus of the present work, often occur together with other side effects of this kind of surgery, because of this it is impossible to establish strict protocols. (Answered in the text and red-highlighted)
